# McCune-Albright Syndrome in Infant with Growth Hormone Excess

**DOI:** 10.3390/genes13081345

**Published:** 2022-07-27

**Authors:** Katarina Brzica, Marko Simunovic, Matea Ivancic, Darija Tudor, Ivna Skrabic, Veselin Skrabic

**Affiliations:** 1Department of Pediatrics, University Hospital of Split, Spinciceva 1, 21000 Split, Croatia; katarina.brzica00@gmail.com (K.B.); darija_tudor@hotmail.com (D.T.); ivna595@gmail.com (I.S.); vskrabic@kbsplit.hr (V.S.); 2Department of Pediatrics, University of Split School of Medicine, Soltanska 2, 21000 Split, Croatia; 3Department of Pediatrics, Sibenik General Hospital, Stjepana Radica 83, 22000 Sibenik, Croatia; matea.ivancic.st@gmail.com

**Keywords:** McCune-Albright syndrome, growth hormone excess, cafe au lait macules, octreotide

## Abstract

Background: McCune-Albright is a rare syndrome, caused by mutation of the *GNAS1* gene, and is characterized by an appearance of multiple endocrinopathies, most commonly premature puberty, polyostotic fibrous dysplasia and skin changes called *cafe au lait* macules. Case report: We present the case of a patient who is, to the best of our knowledge and after extensive review of literature, the youngest McCune-Albright syndrome patient with growth hormone excess, diagnosed at 8.9 months of age. An extensive diagnostic procedure was done upon the diagnosis. Hormonal assessment was performed and all hormone levels were within reference range, and an additional oral glucose suppression that noted the presence of growth hormone excess. Magnetic resonance imaging of the pituitary gland did not detect a tumor process. The genetic analysis of the *GNAS1* gene from skin punch biopsy came back negative. Octreotide was administered as therapy for growth hormone excess at 9.8 months. After the introduction of therapy, we noted a decrease in growth rate from 29.38 to 16.6 cm/year. Conclusion: This case report emphasizes the lack of available data on treatment of growth hormone excess and follow-up in pediatric population and the need for further research.

## 1. Introduction

McCune-Albright syndrome (MAS) is characterized by an appearance of multiple endocrinopathies, most commonly premature puberty, polyostotic fibrous dysplasia causing deformities and pain of limbs, spine and face, accompanied by skin changes in the form of *cafe au lait* macules [1,2,3]. The syndrome is rare, with prevalence estimated at 1 in 100,000 to 1 in 100,000,000 [2]. It is caused by activating mutation of the Gsα subunit of the *GNAS1* gene, located on chromosome 20q13.11, resulting in expression of an activated Gs protein and subsequent overproduction of cyclic adenosine monophosphate (cAMP), which is thought to stimulate the growth and function of osteoblasts, melanocytes and endocrine glands [4,5,6,7]. The mutation is postzygotic, leading to somatic mosaicism in affected endocrine glands, bone and various tissues, with variability of clinical presentation. The distribution of cells containing these mutations is determined by the stage of embryonic development at which mutation occurred [6]. Owing to this, it is difficult to detect the mutation, and a negative result of gene analysis does not exclude the diagnosis. The diagnosis is usually based on the clinical picture [2,4,6,8]. MAS is not considered to be inherited [6].

Endocrinopathies are caused by a stimulation of endocrine glands through autonomic activation of adrenocorticotropic hormone (ACTH), thyroid-stimulating hormone (TSH), luteinizing hormone (LH), follicule-stimulating hormone (FSH) receptors and increased production of targeted hormones, with activation of melanocytes and osteoblasts as well [2,4,9]. Premature puberty is the most common endocrinopathy, it is gonadotropin hormone–releasing hormone (GnRH) independent, with suppressed LH and FSH and elevated levels of testosterone and estradiol [10]. Other endocrinopathies include hyperthyroidism and Cushing’s syndrome [3,11,12]. Diagnosis of growth hormone excess is made by failure to suppress growth hormone in the oral glucose suppression test (OGTT) below 2 mU/L [13]. *Cafe au lait* macules are often seen already in infancy, usually distributed along the central line of the body [3]. 

The prognosis of the syndrome is generally good, but deformities, fractures, compression of cranial nerves and multiple endocrinological complications are possible. It worthy to note that malignant transformation of the affected tissues is possible—neoplasms of the thyroid gland, breast, and bone have been reported [14,15,16].

This case report describes the clinical presentation of McCune-Albright syndrome while noting the risk of development of multiple endocrinopathies and other multisystem complications, together with a diagnostic and therapeutic approach to growth hormone excess in early infancy. We emphasize the lack of available data on treatment of growth hormone excess and follow-up in pediatric population and the need for further research.

## 2. Results

We present the case of a patient, a female toddler, who was diagnosed with McCune-Albright syndrome with growth hormone excess at the age of 8.9 months and is, to the best of our knowledge and after extensive review of literature, the youngest published patient. Informed written consent was obtained from the parents of the patient.

### 2.1. Anamnestic Data and Clinical Examination

She was born from normal pregnancy and birth, with birth weight 4450 g (99. ct., SDS 2.4), birth length 56 cm (>99. ct., SDS 3.68) and head circumference 34.5 cm (70. ct., 0.68 SDS). At the 2 months of age, an accelerated linear increase in body length was observed. Since the age of 3 months she was under supervision of a neuropediatrician and a physical therapist because of hypertonus. At 6 months of age, a *cafe au lait* macule appeared on skin located on the left lumbar (Figure 1), and the growth curve was still accelerated (Figure 2). 

After a video consultation, the infant was referred to a pediatric endocrinologist for examination. At the first examination at the age of 8.9 months, she had a body length of 80 cm (>99. pc., +4.32 SDS according to WHO), body weight 9.67 kg, BMI 15.1 kg/m^2^ (−1.2 SDS according to WHO) (Figure 2).

On the skin, located at the left lumbar region with spreading to the medial line was a *cafe au lait* macule, sized 26 × 7.5 cm (Figure 1). Stigmas were noted: mild asymmetry of the face, hypertelorism, wider root of the nose, longer filtrum, triangular mouth, high palate, wider-spaced mamillas, smaller umbilical hernia. Breasts were Tanner 1, genitals were female, Tanner 1. Bone age was accelerated and estimated at 1.5 years on atlas according to Greulich and Pyle. The patient’s father’s height was 196 cm, mother’s height 165 cm, and the estimated height according to genetic potential, mean parental height (MPH), was 170.1 cm (+1.05 SDS, delta between height at this time and estimated height being +3.27 SDS).

### 2.2. Assessment of Hormonal Status

Detailed hormonal tests were done. Levels of insulin-like growth factor 1 (IGF-1) and insulin-like growth factor-binding protein 3 (IGFBP-3) were normal as were the levels of all the hormones measured: ACTH and cortisol, prolactin, parathyroid hormone (PTH) FSH, LH and estradiol (Table 1). Alkaline phosphatase (ALP) level was normal as well. 

In order to prove the excess of growth hormone, an oral glucose suppression test was performed, which confirmed the hypersecretion of growth hormone, with the highest recorded level of growth hormone being 10.3 mU/L, and the lowest recorded level 2.82 mU/L (Table 2).

### 2.3. Diagnostic Imaging

At the age of 10.23 months, magnetic resonance imaging of the pituitary gland was performed on a 1.5 Tesla device that described a normal pituitary gland without signs of a possible tumor process with a 1.1 cm cyst of the pineal gland as an accidental finding. 

So far, bone scintigraphy has not been performed because the patient does not show signs of bone dysplasia. She does show slight dysmorphic features of the face, which can be attributed to growth hormone excess, but is without bone pain and deformities.

### 2.4. Genetic Analysis

In order to prove the mutation of the *GNAS1* gene, a skin punch biopsy was performed in the area of the *cafe au lait* spot with the aim of isolating genomic DNA (gDNA) from a fibroblast cell culture. The specimen was sent to Invitae Corporation (San Francisco, CA, United States of America).

Genomic DNA obtained from the submitted sample was enriched for targeted regions using a hybridization-based protocol, and sequenced using Illumina technology. Targeted regions were sequenced with ≥50× depth or supplemented with additional analysis. Reads were aligned to a reference sequence (GRCh37), and sequence changes were identified and interpreted in the context of a single clinically relevant transcript (NM_000516.5). Enrichment and analysis focused on the coding sequence of the indicated transcript, 20 bp of flanking intronic sequence, and other specific genomic regions demonstrated to be causative of this disease. Promoters, untranslated regions and other noncoding regions were not interrogated. The result came back negative. 

### 2.5. Treatment

A long-acting somatostatin analogue octreotide was first administered at 9.7 months of age at a dose of 0.6 mg/kg (with body length of 80 cm, +4.32 SDS according to WHO) and is now applied intramuscularly once every 28 days.

### 2.6. Hormonal Status Follow-Up

Hormonal status is regularly monitored. At 9.7 months, elevated prolactin levels were noted (at 903 mIU/L), while IGF-1 and ALP were normal. At 1.39 years, there was an elevated ACTH level (22.8 pmol/L) with normal cortisol (517 nmol/L), possibly due to stress during venipuncture. The IGF-1 and prolactin levels were normal, as were levels of all other hormones measured (LH, FSH, estradiol, PTH, TSH, T4). At 2 years of age, all levels of hormones were in the normal range (Table 1).

### 2.7. Growth Rate Follow-Up

In addition to hormonal status, anthropometric measures together with growth rate are monitored at regular check-ups in order to monitor the effect of therapy. Growth rate from birth to initiation of octreotide therapy at 9.7 months was 29.38 cm/year. After the introduction of therapy, we noted a decrease in growth rate to 16.6 cm/year.

## 3. Discussion

McCune-Albright syndrome has a very variable clinical picture and a wide range of presentations. It usually consists of a triad of premature puberty, polyostotic fibrous dysplasia and *cafe au lait* macules, but several other endocrinopathies and involvement of other tissues have been reported in the literature. Our patient was diagnosed with growth hormone excess at the age of 8.9 months and is, to the best of our knowledge and after extensive review of literature, the youngest patient with McCune-Albright syndrome with growth hormone excess. IGF-1 levels were normal at all measurements, but in the oral glucose suppression test growth hormone levels were not suppressed below 2.82 mU/L, therefore confirming the growth hormone excess. The proportion of patients who have an excess of growth hormone varies in multiple studies, ranging from 15% to over 40% [3,7,8,17,18]. The age at diagnosis is higher than in our patient, with patients usually diagnosed in the second decade of life, while one study described an average age of diagnosis at 6.5 years [8,17,19,20]. In most studies, levels of IGF-1 were elevated. Very rarely IGF-1 was in the reference range, and the diagnosis was made only when it was not possible to suppress growth hormone levels in OGTT [8,17,19]. IGF-1 is the regulator of growth and controls bone elongation; it has an anabolic effect and also promotes the linear growth effect of GH, working in synergy with GH, while IGF-1 binding proteins mediate the activities of IGF-1 [21,22,23]. There are reports that also indicate that there is local IGF-1 production in nonhepatic tissues, such as bones, suggesting that IGF-1 has an endocrine, autocrine and paracrine growth regulatory action independent of GH [22,23,24]. Furthermore, there are also studies supporting the direct effect of GH on the growth plate independent of IGF-1, suggesting that growth hormone directly stimulates the cells in the growth plate through local mechanisms without a rise in circulating IGF-1 [23,24,25,26,27]. However, long-term follow-up of our patient will further clarify whether IGF-1 dependent growth is occurring or not.

No pituitary tumor was observed in our patient on an MRI scan. In the literature, we notice that the share of patients in whom a pituitary tumor is described varies from 33% to over 60%, and diffuse pituitary disease is possible as well [8,17,18,20].

At the age of 9.7 months, a long-acting somatostatin analogue, octreotide, was administered at a dose of 0.6 mg/kg every 4 weeks. Very few data are available in the literature on the proper dosage for the pediatric population for the purpose of suppressing excess growth hormone and on its effect on slowing the growth rate [28,29]. Before the introduction of therapy, the growth rate was 29.38 cm/year; afterward, a gradual decline in growth rate was noted, at 16.6 cm/year. In the treatment of excess growth hormone, medicament therapy is most commonly used; the therapeutic choice consists of long-acting somatostatin analogues (octreotide, lantreotide), long-acting dopamine agonists (carbegolin) and growth hormone receptor antagonists (pegvisomant) [17]. The success of such therapy in normalizing growth hormone levels and IGF-1 has been noted in the literature, although data on its effect on growth rate in the pediatric population is very limited. According to available literature, octreotide was successful in normalizing IGF-1 levels, either fully or partially [17,20,30]. The response to pegvisomant has also been shown to be satisfactory in lowering growth hormone levels and IGF-1 levels, in combination with octreotide or as sole therapy [8,20,30]. In several case reports, pegvisomant in children has been shown to reduce IGF-1/IGFBP-3 levels and partially normalize growth rate [20,31,32,33]. The response to the dopamine agonist cabergoline in the normalization of IGF-1 and GH is somewhat weaker [8,30]. As IGF-1 levels are normal in our patients, it is difficult to compare it to available literature. Other therapeutic modalities are, in case of a pituitary tumor, surgery or radiotherapy [17].

During the one of the follow-ups, the level of prolactin was slightly elevated, while in other measurements the level was in reference range. In patients with McCune-Albright syndrome with excess growth hormone, depending on the study, a level of elevated prolactin cosecretion was observed in 46.1% to 92% of patients [8,17,18,19,20]. The reason for this elevated cosecretion is unclear; the explanation may lie in the existence of pituitary adenoma of mixed cellularity or in the disorder of differentiation of cells and, consequently, the secretion of both hormones [7,8,18,34]. Patients with MAS may have hyperthyroidism; the frequency varies from 14.3% to 30.8% [17,18,19]. Cushing’s syndrome is rarely reported, it usually occurs in neonatal period [3,11,12,35]. 

At first examination, a mild asymmetry of the face was noted and it recently became slightly more noticeable, which can be attributed to sole excess of growth hormone, but also signifies possible beginning of fibrous dysplasia of the bones [7]. Studies have described higher frequency of associated disorders in patients with excess growth hormone, primarily fibrous dysplasia, but also impairment of vision, hearing and smell [7,8,17,20]. It has also been observed that uncontrolled excess of growth hormone exacerbates fibrous dysplasia [18,19]. Owing to increased bone replacement, ALP is elevated in some patients, and the level correlates with bone involvement. ALP levels normalize with good control of growth hormone suppression [17,19]. The level of ALP in our patient was normal at each measurement.

Detection of the *GNAS1* mutation is used to make a definitive diagnosis. This test has its limitations—in the DNA of peripheral blood leukocytes, mutation is found in a small number of patients, on average in about 21% to 45% with variability depending on the number of symptoms. In the biopsy of the affected tissue, this percentage increases by up to 90%, but such a test is very invasive and is rarely done. The exception is biopsy of *cafes au lait* macules, which is positive in up to 50% of patients [9].

## 4. Conclusions

In this case report, to the best of our knowledge, we described the youngest patient diagnosed with McCune-Albright syndrome with excess growth hormone. It should be noted that levels of growth hormone, IGF-1 and IGFBP-3, were normal at each measurement. After confirmation of excess growth hormone level by OGTT, octreotide was administered. We encountered a challenge in introducing the therapy because the available literature contains very sparse information on the use of this drug for the purpose of suppressing excess growth hormone in the pediatric population. Data on recommended doses and monitoring the effect of therapy are lacking. We presented data on the clinical course in the first year of treatment follow-up; we note that we did not achieve satisfactory suppression of growth rate. The *GNAS1* mutation was not detected by DNA isolation in skin biopsy of the *cafe au lait* macule, but a negative result does not exclude the diagnosis because the probability of detecting this mutation from skin change is low. Because of the possibility of development of other endocrinopathies and fibrous dysplasia, our patient will be under long-term surveillance.

## Figures and Tables

**Figure 1 genes-13-01345-f001:**
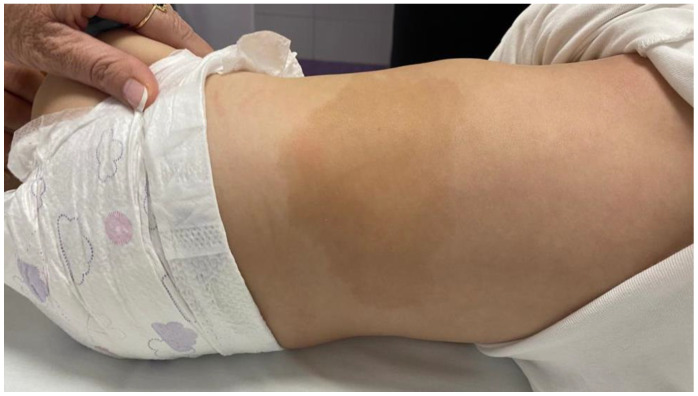
*Cafe au lait* macule.

**Figure 2 genes-13-01345-f002:**
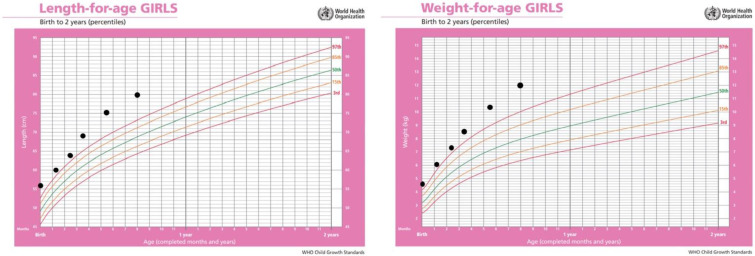
Body length and weight percentile of the patient before first examination by the pediatric endocrinologist, shown on World Health Organization growth charts.

**Table 1 genes-13-01345-t001:** Levels of measured hormones at outpatient controls.

Age at Measurement	8.9 Months	9.1 Months	1.39 Years	2 Years	Normal Range
IGF-1 (nmol/L)	8.7 (ref. 2.34–18.98)	8.32 (ref. 2.34–18.98)	5.52 (ref. 2.6–20.67)	5.02 (ref. 2.6–20.67)	depending on age
IGFBP-3 (nmol/L)	2.73	/	/	/	depending on age
FSH (IU/L)	5.89	/	8.04	10.6	0.2–11.1
LH (IU/L)	<0.3	/	<0.3	0.312	0.312
E2 (pmol/L)	<18.4	/	<18.4	<18.4	<37.7
PRL (mIU/L)	300	903	427	303	102–496
ACTH (pmol/L)	11.6	/	22.8	6.5	1.6–13.9
cortisol (nmol/L)	288	/	517	224	171–536
TSH (mIU/L)	/	/	1.36	1.87	0.7–5.97
T4 (nmol/L)	/	/	80.6	108	76.6–189
PTH (pmol/L)	3.4	/	4.2	/	1.58–6.03

IGF-1, insulin-like growth factor 1; IGFBP-3, insulin-like growth-factor-binding protein 3; FSH, follicule-stimulating hormone; LH luteinizing hormone; E2 estradiol; PRL, prolactin; ACTH adrenocorticotropic hormone; TSH, thyroid-stimulating hormone; T4, thyroxine; PTH, parathyroid hormone.

**Table 2 genes-13-01345-t002:** Oral glucose suppression test results.

Time (Minutes)	0	30	60	90	120
Glucose (mmol/L)	4.23	5.55	4.73	5.25	3.64
Insulin (uU/mL)	2.4	10.8	5.9	8.8	1.2
Growth hormone (mU/L)	6.45	2.82	3.64	3.33	10.3

## Data Availability

The data presented in this study are available on request from the corresponding author.

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
