# Peer review of "McCune-Albright Syndrome in Infant with Growth Hormone Excess"

_genes, 2022, doi:10.3390/genes13081345_

Round 1

Reviewer 1 Report

The authors describe a case of Mc Cune-Albright syndrome (MAS) with GH excess, diagnosed in a female infant at the age of 8.9 months.  She had shown an accelerated linear increase in body length from 2 months of age  onwards. Diagnosis of MAS was made on the basis of clinical  examination evidencing  some  characteristic features of this syndrome even if the genetic analysis searching for mutation of GNAS1 gene by skin biopsy in the area of cafe au lait spot did not evidence any mutation. Hormonal pattern, including IGF1 and IGFBP-3 , was normal and GH excess was diagnosed on the basis of the response to glucose tolerance test. MRI did not evidence pituitary adenoma, Thus, an octreotide therapy was started at 9.8 months and induced a significant decrease in growth rate.

COMMENT. GH excess in MAS has already been described  , however this  paper is of some interest due to the youngest  age of this patient with respect to other patients with MAS and GH excess so far described in the literature.                                                                                                    Major concerns.                                                                                                 1.The authors should discuss in more detail the mechanisms that caused the rapid increase in body lenght in this patient, who had normal levels of IGF1 and its ligand IGFBP-3, but also of alkaline phosphatase. The GH action in promoting linear body growth is  mostly mediated by IGF1, that stimulates the epiphysial cartilage to grow and only minimally by the utilization of spared proteins derived by the metabolic effect of GH. Thus, the rapid increase  of linear growth rate in few months of lif in this patient is difficult to explain without the involvement of IGF1 increase.                                             2. A possible therapy with Pegvisomant, a GH receptor antagonist, should be explored in comparison to octreotide.

Minor concern

Also the review by Christofondis et al. (The McCune-Albright syndrome: growth hormone and prolactin hypersecretion.J Pediatr Endocrinol Metab ,2006) shoud be discussed in the text and cited in bibliography.

Author Response

  1. We would like to address the following comment:

The authors should discuss in more detail the mechanisms that caused the rapid increase in body lenght in this patient, who had normal levels of IGF1 and its ligand IGFBP-3, but also of alkaline phosphatase. The GH action in promoting linear body growth is mostly mediated by IGF1, that stimulates the epiphysial cartilage to grow and only minimally by the utilization of spared proteins derived by the metabolic effect of GH. Thus, the rapid increase of linear growth rate in few months of lif in this patient is difficult to explain without the involvement of IGF1 increase.“

Comment: Following the Reviewer’s comment, and after extensive review of literature, we did our best to give a possible explanation for accelerated height velocity with normal circulating IGF-1 levels and we expanded Discussion section of our manuscript.

  1. We would like to address the following comment:

“A possible therapy with Pegvisomant, a GH receptor antagonist, should be explored in comparison to octreotide.”

Comment: Following the Reviewer’s comment, we added a paragraph exploring different therapeutic options, including pegvisomant, in our manuscript.

  1. We would like to address the following comment:

“Also the review by Christofondis et al. (The McCune-Albright syndrome: growth hormone and prolactin hypersecretion.J Pediatr Endocrinol Metab ,2006) shoud be discussed in the text and cited in bibliography.”

Comment: Following the Reviewer’s comment, we have discussed and cited the review mentioned above.

Reviewer 2 Report

It's a very interesting case report

- I Think that the discussion section has some paragraph identical to introduction. I could be rewritten

- Did the author performed bone age of this child before and after treatment?

- Did the child undergo bone scintigraphy to assess bone displasia?

- At the discussion section, page 5, line 176 and 182: I did not understand these different frequency of patients with MAS and GH excess

Author Response

  1. We would like to address the following comment:

“I Think that the discussion section has some paragraph identical to introduction. I could be rewritten”

Comment: Following the Reviewer’s comment, we have reviewed the mentioned paragraphs and rewritten parts that were similar.

  1. We would like to address the following comment:

“Did the author performed bone age of this child before and after treatment?”

Comment: Following the Reviewer’s comment, we would like to point out that bone age has not been assessed again during therapy because of the short period of admission of octreotide, we planned to do it annually as provided by the standard protocols for the reduction of ionizing radiation with regard to the patient's age.

  1. We would like to address the following comment:

“Did the child undergo bone scintigraphy to assess bone displasia?”

Comment: Following the Reviewer’s comment, we addressed this question in our manuscript, in section 2.3 Diagnostic imaging, on page 5. We would like to point out that scintigraphy was planned in near future.

  1. We would like to address the following comment:

“At the discussion section, page 5, line 176 and 182: I did not understand these different frequency of patients with MAS and GH excess.”

Comment: Following the Reviewer’s comment, we reviewed the section mentioned and elaborated it more clearly. 

Round 2

Reviewer 1 Report

I think the paper sufficiently improved to warrant publication in Gene after minor revision of the English